·◌· PLOS | ONE

# Autoantibody production significantly decreased with APRIL/BLyS blockade in murine chronic rejection kidney transplant model

Natalie M. Bath[1], Xiang Ding[1], Bret M. Verhoven[1], Nancy A. Wilson[2], Lauren Coons[1], Adarsh Sukhwal[1], Weixiong Zhong[3], Robert R. Redfield III[1] *

1 Department of Surgery, Division of Transplant, University of Wisconsin-Madison, Madison, Wisconsin, United States of America, 2 Department of Medicine, Division of Nephrology, University of Wisconsin-Madison, Madison, Wisconsin, Unites States of America, 3 Department of Pathology, University of Wisconsin–Madison, Madison, Wisconsin, United States of America

* redfield@surgery.wisc.edu

**Data Availability Statement:** All relevant data are within the manuscript.

## Abstract

Chronic antibody mediated rejection (cAMR) remains a significant barrier to achieving long-term graft survival in kidney transplantation, which results from alloantibody production from B lymphocytes and plasma cells. APRIL (A proliferation-inducing ligand) and BLyS (B lymphocyte stimulator) are critical survival factors for B lymphocytes and plasma cells. Here we describe the results of APRIL/BLyS blockade in a murine cAMR kidney transplant model. c57/B6 mice underwent kidney transplantation with Bm12 kidneys (minor MHC mismatch), a well-described model for chronic rejection where animals cannot make donor specific antibody but rather make antinuclear antibody (ANA). Following transplantation, animals received TACI-Ig (to block APRIL and BLyS) or no treatment. Animals were continued on treatment until harvest 4 weeks following transplant. Serum was analyzed for circulating anti-nuclear autoantibodies using HEp-2 indirect immunofluorescence. Spleen and transplanted kidneys were analyzed via H&E. ANA production was significantly decreased in APRIL/BLyS blockade treated animals (p<0.0001). No significant difference in autoantibody production was found between syngeneic transplant control (B6 to B6) and APRIL/BLyS blockade treated animals (p = 0.90). Additionally, disruption of splenic germinal center architecture was noted in the APRIL/BLyS blockade treated animals. Despite the significant decrease in autoantibody production and germinal center disruption, no significant difference in lymphocyte infiltration was noted in the transplanted kidney. APRIL/BLyS blockade resulted in a significant decrease of autoantibody production and disrupted splenic germinal center formation in a chronic kidney transplant model, however in this model no difference in kidney transplant pathology was seen, which may have to do with the absence of any T cell centric immunosuppression. Regardless, these findings suggest that APRIL/BLyS blockade may play a role in decreasing antibody formation long-term in kidney transplantation. Future investigations will use APRIL/BLyS blockade in conjunction with T lymphocyte depleting agents to determine its efficacy in chronic rejection.

**Funding:** TACI-Ig was kindly provided by EMD Serono Research and Development Institute under an MTA. This study was supported by KL2 grant KL2TR002374, grant UL1TR002373 to UW ICTR, and UW Transplant Research Training Grant (T32 AI125231) from NIH/NCATS. Dr. Redfield's research was also supported in part by a fellowship from the American College of Surgeons, and a Faculty Development Grant from the American Society of Transplant Surgeons Foundation. The UW CCC Flow Cytometry Shared instrumentation core, including the Shared Instrumentation grant 1S00OD018202-01 Special BD LSR Fortessa, made possible the purchase and use of the BD LSR Fortessa. The funders had no role in study design, data collection and analysis, or decision to publish.

**Competing interests:** We have the following interests. This study was partly funded by EMD Serono Research and Development Institute under an MTA, and TACI-Ig was kindly provided by EMD Serono Research and Development Institute under an MTA. There are no patents, products in development or marketed products to declare. This does not alter our adherence to all the PLOS ONE policies on sharing data and materials, as detailed online in the guide for authors.

## Introduction

Antibody mediated rejection (AMR) is widely recognized as a common cause for late kidney allograft failure.[1–3] Mature B lymphocytes and plasma cells (terminally differentiated B lymphocytes) play a critical role not only in the development of AMR through donor specific antibody (DSA) production, but they also function as effector cells in T lymphocyte activation, which may result in cellular rejection.[4, 5] Current strategies used to treat AMR include anti-CD20 antibodies (rituximab), antibody removal (plasmapheresis, intravenous immunoglobulin), and proteasome inhibitors to target plasma cells, which are the producers of alloantibody. [6, 7] Despite these efforts, chronic antibody mediated rejection (cAMR) remains a significant barrier to achieving long-term graft survival in kidney transplantation. As a result, novel and effective strategies to treat antibody mediated rejection remain an unmet need.[8]

Due to the multiple functions of B lymphocytes as both effector and alloantibody-producing cells, it is our hypothesis that B lymphocytes will need to be targeted at various stages of development in order to successfully prevent DSA production and rejection. APRIL (A proliferation-inducing ligand) (TNFSF 13a) and BLyS (B lymphocyte stimulator) (TNFSF 13b) are critical survival factors for plasma cells and B lymphocytes, respectively. APRIL plays an important role in plasmablast and plasma cell survival.[9, 10] BLyS acts as an important co-stimulator of B lymphocyte survival and proliferation.[4, 11, 12] APRIL and BLyS have been individually investigated as potential therapeutic targets in oncology and transplant clinical trials, respectively. A recent randomized-controlled trial investigated the efficacy of anti-BLyS antibody (belimumab, GlaxoSmithKline) in addition to standard-of-care immunosuppression to reduce naïve B cells. Although this endpoint was not met, belimumab reduced memory B cells, which suggests it may have a role in long-term desensitization strategies in patients with pre-existing DSA.[13] B cell maturation antigen (BCMA), which APRIL and BLyS bind to on the surface of memory B and plasma cells, has been targeted in multiple myeloma clinical trials using anti-BCMA antibody with promising results.[14, 15]

We have previously reported the significant B lymphocyte depletion and DSA reduction that occurs with APRIL/BLyS blockade in allosensitized mice.[16] However, APRIL/BLyS blockade did not result in any difference in acute rejection rates, which may be due to a need for extended treatment. Therefore, our goal of the current study was to determine the effect of targeting both APRIL and BLyS in a chronic rejection kidney transplant model. Previously, MHC class II-mismatched Bm12 (B6.H-2$^{bm12}$) to B6 (c57/BL6) has been described as a murine model of chronic heart rejection but has not been applied to murine kidney transplant models. [17–19] Here we describe the results of APRIL/BLyS blockade in a novel murine chronic rejection kidney transplant model.

## Materials and methods

### Animals

C57BL/6 (H-2$^b$) and B6.H-2$^{bm12}$ (H-2$^{bm12}$) mice were purchased from Jackson Laboratories (Bar Harbor, ME) and housed in the University of Wisconsin Laboratory Animal Facility. All procedures were performed in accordance with the Animal Care and Use Policies at University of Wisconsin. Animal health including animal deaths, room temperature, 12-hour light/dark cycles, and cage cleaning among other sanitation duties were performed daily by WIMR housing staff. Food and water were available ad libitum. This research was prospectively approved by School of Medicine and Public Health Institutional Animal Care and Use Committee at the University of Wisconsin (M005204). Animals that underwent transplantation were monitored daily post-transplant. Animal health was evaluated by activity level, weight gain or loss,

hunched posture, and other signs of distress. C57BL/6 animals were transplanted with a Bm12 kidney with simultaneous nephrectomy as previously described.[20] Animals were then randomized to no treatment or APRIL/BLyS blockade (treated with TACI-Ig (Transmembrane activator and calcium modulator and cyclophilin ligand interactor-Immunoglobulin) (100 μg TACI-Ig in PBS, i.p. injection 3x/week for 28d)) post-transplantation. TACI-Ig blocked both APRIL and BLyS. Animals were anesthetized with isoflurane during surgery or injections and sacrificed via cardiac puncture. Buprenorphine (1mg/kg SQ) was used administered post-transplant and every 72 hours for the first week following surgery. C57BL/6 animals transplanted with a c57BL/6 kidney were used as syngeneic transplant control. The contralateral native kidney was removed on day 1 post-transplant, leaving the animal to rely fully on the transplanted kidney. Spleen, blood, urine, and kidney were collected at 28d post-transplant for immediate utilization, storage in 10% formalin for immunohistochemistry (IHC), or were processed to single cells and cryopreserved in liquid nitrogen.

## Quantification of circulating autoantibodies and biochemical

Serum samples were collected from mice 1 month post-transplant at the time of harvest. Circulating autoantibody levels were determined using NOVA Lite HEp-2 indirect immunofluorescent assay. Briefly, the serum was diluted to 1:40 with PBS, incubated with antigen substrate slides and unreacted antibodies are washed off. The substrate was then incubated with anti-mouse IgG FITC and the unbound reagent was washed off. A fluorescent microscope was used to image autoantibody positive samples. Quantification was performed using a custom macro written for ImageJ software (NIH, imagej.nih.gov/ij/). Three to five non-overlapping pictures of representative images were taken from animals for ImageJ analysis.

Urine protein and creatinine was measured on an IdexxVetTest 8008 bioanalyzer (Idexx Laboratories, West Sacramento, CA) using compatible assay chips according to manufacturer's instructions. Proteinuria severity was calculated using the urine protein to creatinine (UPC) ratio and graded as none (UPC <0.5), mild (UPC 0.5–1.0), moderate (UPC 1.0–2.0), and severe (UPC >2.0).

## Flow cytometry

Single cell suspensions of splenocytes were prepared from fresh cells. Flow methods were similar to Allman and Gross.[21, 22] After Ficoll purification, splenocytes underwent ACK lysis of red blood cells. After counting and re-suspension in R10 (RPMI with 10% Fetal Calf Serum), 500,000 cells were added to cluster tubes and stained. Cells from each tissue were stained for B lymphocyte subsets, T lymphocyte subsets, and regulatory T cells (Tregs). Antibodies used for B lymphocyte subsets include: Alexa Fluor 488 anti-mouse IgD (11-26c.2a BioLegend), PerCP rat anti-mouse CD45R/B220 (RA3-6B2 BD Pharmingen), PE rat anti-mouse CD24 (M1/69 BD Pharmingen), PE/Cy7 rat anti-mouse IgM (R6-60.2 BD Pharmingen), BV421 rat anti-mouse CD3 (17A2 BioLegend), BV605 hamster anti-mouse CD27 (LG.3A10 BD Horizon), BV711 rat anti-mouse CD38 (90/CD38 BD OptiBuild), FITC rat anti-mouse CD21/CD35 (7G6 BD Pharmingen), PE rat anti-mouse CD5 (clone 53–7.3 BD Pharmingen), APC anti-mouse CD23 (B3B4 BioLegend), and viability dye Ghost Dye Red 780 (13-0865-T100 Tonbo Biosciences). Antibodies used for T lymphocyte subsets and Tregs include: PE rat anti-mouse CD25 (PC61 BD Pharmingen), PE/Cy5 rat anti-mouse CD4 (GK1.5 BioLegend), BV605 rat anti-mouse CD8α (53–6.7 BD Horizon), Alexa Fluor 647 anti- mouse FOXP3 (150 D BioLegend), and viability dye Ghost Dye Red 780 (13-0865-T100 Tonbo Biosciences). Flow cytometry was performed on a BD LSR II at the UWCCC Flow Cytometry Laboratory and data analyzed with FlowJo (TreeStar, Inc., Ashland, OR).

**B lymphocyte subset gating.** Cells were gated to remove non-singlets, then gated through a live/dead gate, and lastly through tight lymphocyte gate based on forward and side scatter. $CD3^-$ lymphocytes were gated in and then visualized as IgD versus CD45R. Memory B lymphocytes were defined as $CD3^-CD27^+CD45R^+$ from the $CD3^-$ gate. From the $CD3^-$ lymphocyte gate, mature B ($CD3^-CD21^+IgM^+$), transitional 1 (T1) ($CD3^-CD21^-IgM^{++}$), and transitional 2 marginal zone (T2 MZ) B ($CD3^-CD21^{++}IgM^{++}$) lymphocytes were gated. Additionally, CD45R *versus* CD5 were gated as $CD45R^+CD5^-$ cells then visualized as CD23 *versus* CD21. Gates were drawn for $CD21^-CD23^-$ (newly formed B lymphocytes) and $CD21^{int}CD23^+$ (follicular (Fo) B lymphocytes).

**Plasma cell gating.** Cells were gated to remove non-singlets, then through a large gate to ensure capture of the typically larger plasma cells, and were subsequently visualized in an IgD versus CD45R gate. $IgD^-CD45R^-$ cells were then visualized as CD27 versus IgM. Plasma cells were defined as $CD3^-IgD^-CD45R^-CD27^+IgM^-CD38^+$. Normalized cell counts for plasma cells were calculated based on the large gate, rather than the lymphocyte gate as was used for other lymphocyte populations.

**T lymphocyte subset gating.** Cells were gated to remove non-singlets, then gated through a tight lymphocyte gate based on forward and side scatter. Cells were then visualized as CD4 *versus* CD3 and CD8 *versus* CD3. T lymphocytes were defined as $CD4^+CD3^+$ or $CD8^+CD3^+$. $CD4^+CD3^+$ cells were further gated as CD25 *versus* FOXP3. Tregs were defined as $CD4^+CD3^+CD25^+FOXP3^+$.

## Histology

Kidneys were collected at time of euthanasia and 1/3 of the spleen was preserved in 10% formalin for at least 24 hours, processed, paraffin embedded, and cut into 5 μm sections. After deparaffinizing and rehydrating, spleen sections were stained with anti-PAX5 (Abcam, ab140341 polyclonal). The ImmPRESS HRP reagent and ImmPACT DAB substrate were used to detect anti-PAX5 primary antibody as a brown pigment. Kidney sections were washed in distilled water, counter-stained with hematoxylin and eosin (H&E) or periodic acid-Schiff (PAS) stain and dehydrated through an ethanol series. Slides were imaged on a Nikon Eclipse E600 supplied with an Olympus DP70 camera. Automated quantification was performed using a custom macro written for ImageJ software (NIH, imagej.nih.gov/ij/). All H&E and PAS slides were reviewed by a transplant pathologist (WZ) blinded to study groups and scored for rejection according to Banff 2013 guidelines.[23]

## Statistics

Statistics were performed using the statistical packages that are part of Prism 7 for Windows, v 7.0b. ANOVA, T-tests, and chi-square were primarily used. P values of 0.05 or less were considered significant. Statistical calculations to determine power were determined prior to implementation of this experiment.

## Results

### APRIL/BLyS blockade significantly decreased anti-nuclear autoantibody production

Animals were harvested 4 weeks following transplant and their tissues were collected to evaluate for changes in ANA production, chronic kidney rejection, and B and T lymphocyte populations. This model has previously been established in preclinical cardiac transplant models as a method to study chronic rejection without immunosuppression because all allografts will

develop vasculopathy.[24] ANA was used as a marker for antibody production because the minor MHC mismatch between Bm12 and C57/Bl6 (3 amino acids) does not result in alloantibody production, but rather autoantibody.[25] Post-transplant treatment with APRIL/BLyS blockade resulted in a significant reduction of serum ANA (p<0.0001) (Fig 1). Serum ANA detected in the APRIL/BLyS blockade treated group was not significantly different from that seen in the syngeneic kidney transplant group.

## APRIL/BLyS blockade reduces mature B lymphocytes but does not alter newly formed B lymphocytes

Splenic B lymphocyte populations were evaluated using flow cytometry. Overall, mature B lymphocyte populations, which rely on BLyS for development, were significantly decreased in the APRIL/BLyS blockade treated group. Follicular B cells (CD3$^-$CD21$^{int}$CD23$^+$) were reduced in APRIL/BLyS blockade treated animals compared to both untreated transplant (p<0.008) and syngeneic transplant (p<0.05) (Fig 2A). Newly formed B cells (CD3$^-$CD21$^-$CD23$^-$) remained unchanged between groups (Fig 2B). Additionally, early stages of transitional 1 (T1) B cells were not significantly different between treated and untreated groups (Fig 3A). However, transitional zone B cells in later stages of development were depleted with APRIL/BLyS blockade as seen in the transitional 2 (T2) marginal zone B cell population (CD3$^-$CD21$^{++}$IgM$^+$$^+$) (p<0.009) (Fig 3B). Mature B cells (CD3$^-$CD21$^+$IgM$^+$) were also significantly reduced in the APRIL/BLyS blockade treated group compared to untreated transplant and syngeneic transplant (p<0.004) (Fig 3C).

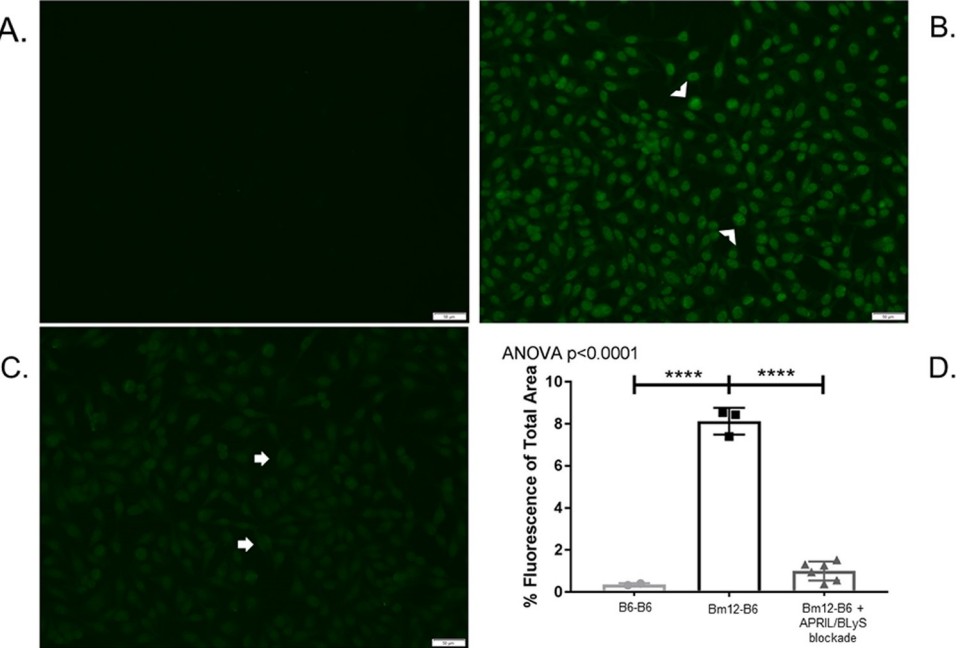

**Fig 1. APRIL/BLyS blockade resulted in significant reduction of anti-nuclear autoantibodies (ANA).** Representative images from indirect immunofluorescence assay detecting circulating anti-nuclear autoantibodies (ANA). (A) B6-B6 transplant group, (B) Bm12-B6 transplant group with no treatment. Arrowheads indicate high-intensity staining. (C) Bm12-B6 transplant group treated with APRIL/BLyS blockade. Arrows indicate low-intensity staining. (D) Densitometry using FITC anti-IgG to detect ANA formation in serum. Graph depicts percentage of total area that was fluorescent. APRIL/BLyS blockade resulted in a significant decrease of total fluorescence indicating a significant decrease in circulating ANA.

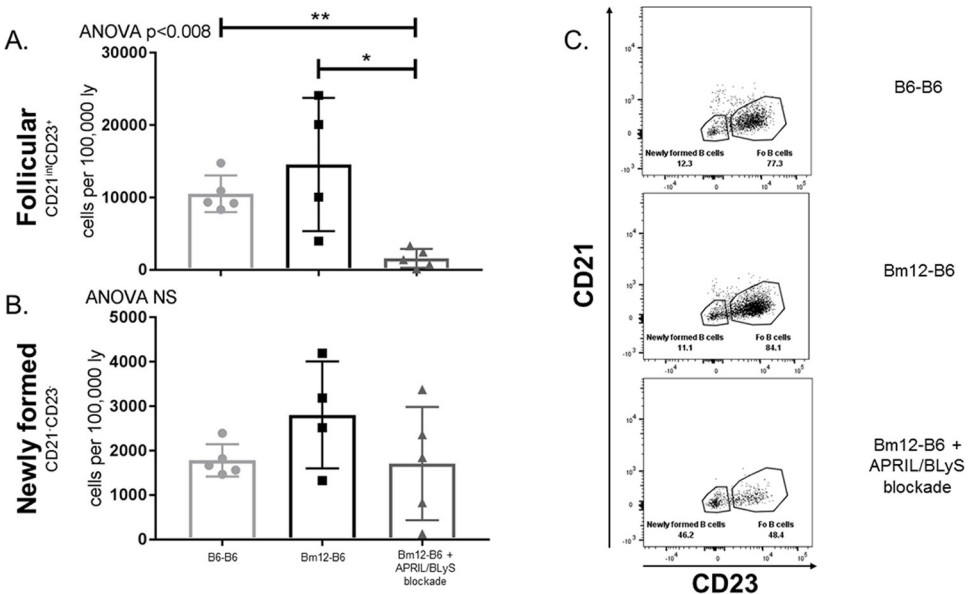

**Fig 2. Follicular B lymphocytes were significantly depleted with APRIL/BLyS blockade, but newly formed B cells remained unchanged.** Flow cytometry was used to assess immature and mature B lymphocyte populations in spleen. Each graph shows number of cells per 100,000 lymphocytes. (A) Follicular B lymphocytes were defined as $CD3^-CD21^{int}CD23^+$. (B) Newly formed B cells were identified as $CD3^-CD21^-CD23^-$. (C) Representative flow cytometry data of follicular and newly formed B cells by group. Number shown represents percentage of total cells in gate. $^*p<0.05$, $^{**}p<0.008$.

## Plasma cell, but not memory B cell, populations declined following treatment with APRIL/BLyS blockade

Memory B and plasma cells were assessed to determine if APRIL/BLyS blockade affected long-lived B lymphocyte populations. Memory B cells ($CD3^-CD27^+CD45R^+$) were unchanged despite treatment with APRIL/BLyS blockade (Fig 4A). Plasma cells ($CD3^-IgD^-CD45R^-CD27^+IgM^-CD38^+$) were significantly decreased in APRIL/BLyS blockade treated group compared to untreated transplant groups ($p<0.04$) (Fig 4B).

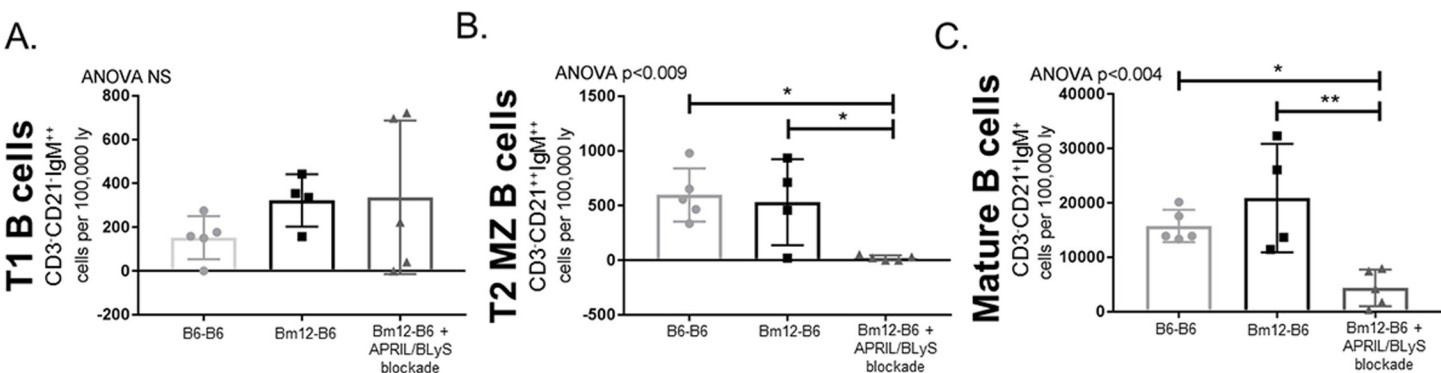

**Fig 3. APRIL/BLyS blockade significantly depleted mature B and transitional 2 (T2) marginal zone B lymphocyte subsets, but preserved immature B lymphocytes.** Flow cytometry was used to assess immature and mature B lymphocyte populations in spleen. Each graph shows number of cells per 100,000 lymphocytes. (A) T1 B cells were defined as $CD3^-CD21^-IgM^{++}$. (B) T2 marginal zone B cells were identified as $CD3^-CD21^{++}IgM^{++}$. (C) Mature B cells were defined as $CD3^-CD21^+IgM^+$. $^*p<0.03$, $^{**}p<0.004$.

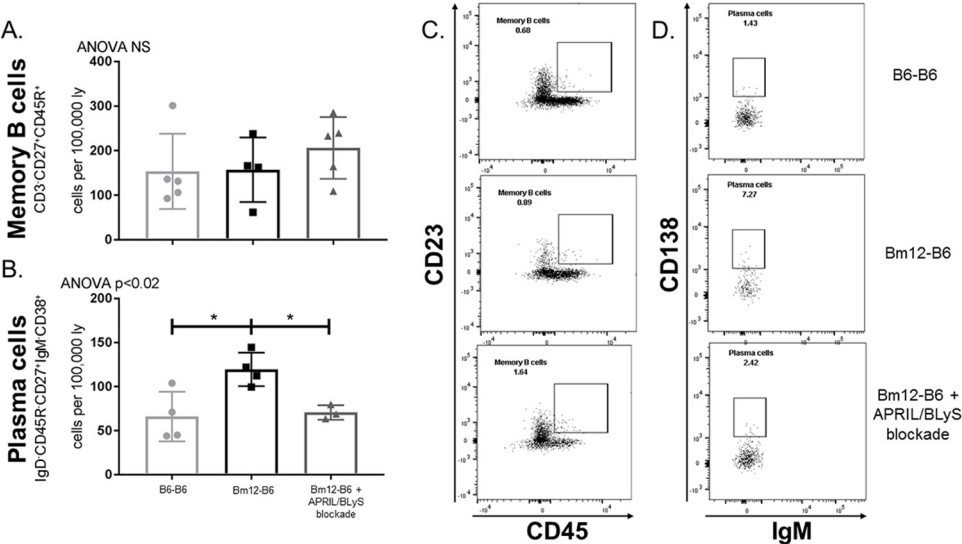

**Fig 4. Plasma cells, but not memory B cells, were significantly decreased with APRIL/BLyS blockade.** Flow cytometry was used to assess long-lived B cell populations in spleen. (A) Graph shows number of cells per 100,000 lymphocytes. Memory B cells were defined as $CD3^-CD27^+CD45R^+$. (B) Graph shows number of plasma cells per 100,000 cells collected. Plasma cells were defined as $CD3^-IgD^-CD45R^-CD27^+IgM^-CD38^+$. (C) Representative flow cytometry data of memory B and plasma cells by group. Number shown represents percentage of total cells in gate. *$p<0.05$.

## Germinal centers in spleen were significantly disrupted in groups treated with APRIL/BLyS blockade

Splenic germinal centers were evaluated via immunohistochemistry using PAX5 antibody in order to further characterize changes in B lymphocyte populations. PAX5 has previously been established as a marker for B lymphocytes.[26] Germinal centers demonstrated normal architecture in syngeneic and untreated groups but was completely disrupted in APRIL/BLyS blockade treated animals (Fig 5). The depletion of B lymphocytes in splenic germinal centers was also confirmed by an overall decrease in anti-PAX5 staining in APRIL/BLyS blockade treated groups compared to other groups ($p<0.0001$).

## APRIL/BLyS blockade results in significantly decreased regulatory T cells (Tregs) but increases effector T cell populations

After characterizing B lymphocytes with APRIL/BLyS blockade, T cell populations were assessed in order to determine if the changes seen in B lymphocyte populations resulted in T cell alterations. Interestingly, both $CD3^+CD4^+$ and $CD3^+CD8^+$ T cell populations were significantly increased in the APRIL/BLyS blockade treated group compared to both syngeneic transplant ($p<0.009$) and untreated transplant groups ($p<0.005$). Conversely, Tregs in untreated and treated transplant groups were significantly lower compared to syngeneic transplant ($p<0.05$) (Fig 6).

## Despite changes in ANA production and B lymphocytes, APRIL/BLyS blockade did not prevent rejection or preserve kidney function

Finally, we evaluated the transplanted kidneys 4 weeks post-transplant. Histology was reviewed by a transplant pathologist (blinded) and scored for acute and chronic peritubular capillaritis (ptc), glomerulitis (g), tubulitis (t), vasculitis (v), interstitial inflammation (i), mi

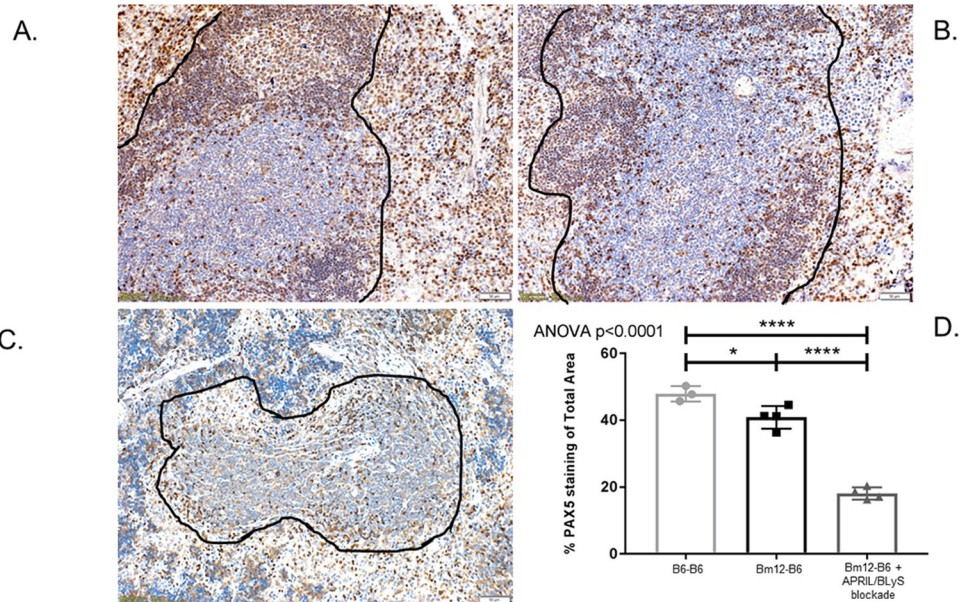

**Fig 5. Spleen PAX5 significantly decreased in APRIL/BLyS blockade treated mice.** Preservation of splenic germinal center seen in (A) syngeneic transplant control and (B) untreated transplant group. B lymphocytes are shown in brown; T lymphocytes are shown in blue. (C) Destruction of normal germinal center seen in animals treated with APRIL/BLyS blockade as indicated black outline. (D) Densitometry using anti-PAX5 antibody to detect B lymphocytes in splenic germinal centers. Graph depicts percentage of total area of spleen staining for anti-PAX5.

(microcirculation inflammation), and arteriolar hyalinosis (ah) according to Banff 2013. Overall, no difference in rates of cellular rejection were seen between untreated and treated transplant groups (Fig 7). The overwhelming level of cellular rejection noted in the treated group is

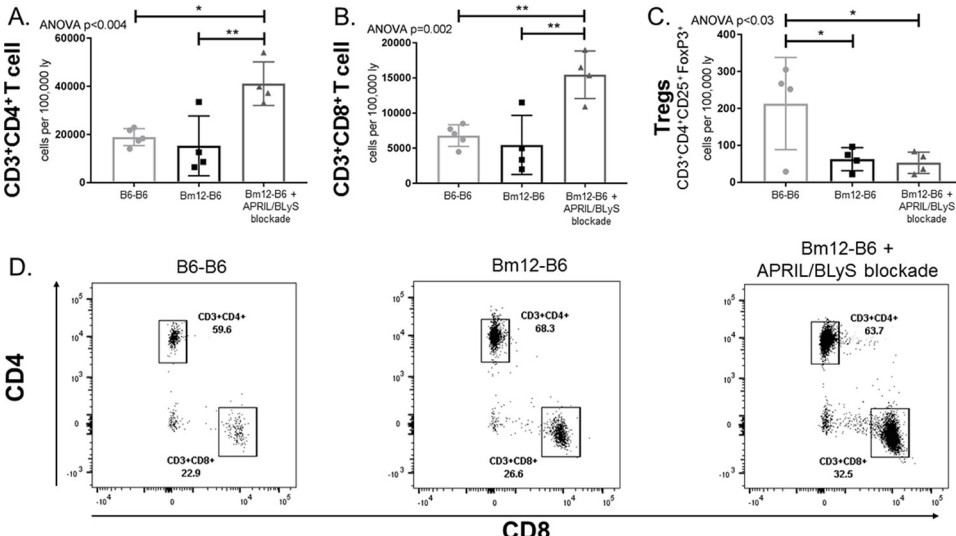

**Fig 6. APRIL/BLyS blockade significantly reduces regulatory T cells (Tregs), but increases effector T cell populations.** Flow cytometry was used to assess T cell populations in spleen. Each graph shows number of cells per 100,000 lymphocytes. (A) CD3+CD4+ T cells. (B) CD3+CD8+ T cells. (C) Regulatory T cells (Tregs) were defined as CD3+CD4+CD25+FoxP3+. (D) Representative flow cytometry data of CD3+CD4+ and CD3+CD8+ from B6-B6 (Left), Bm12-B6 (Middle), and Bm12-B6 + APRIL/BLyS blockade treated group (Right). Number shown represents percentage of total cells in gate. *p<0.05. **p<0.009.

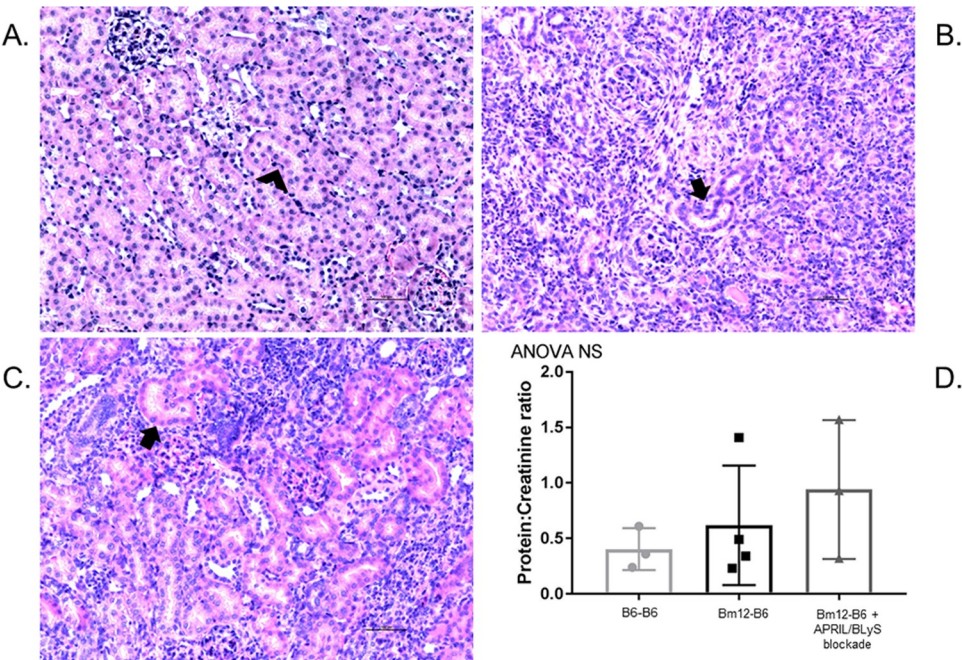

**Fig 7. APRIL/BLyS blockade did not prevent rejection in kidney transplant model.** Animals underwent kidney transplant and were randomized to receive no treatment or APRIL/BLyS blockade for 1 month post-transplant. (A) B6-B6 kidney transplant did not develop any evidence of rejection. Arrowhead indicates normal kidney tubule. (B) Bm12-B6 developed rejection in 100% of animals and 16.7% (N = 1) developed AMR. Arrow indicates damaged tubule. (C) Bm12-B6 animals treated with APRIL/BLyS blockade also overwhelmingly developed ACR. Arrow indicates damaged tubule. (D) Urine protein and creatinine (UPC) was measured to assess kidney function. UPC ratio was scored as none (UPC <0.5), mild (0.5–1.0), moderate (UPC 1.0–2.0), and severe (UPC >2.0) proteinuria. Proteinuria was not improved in animals who received APRIL/BLyS blockade.

consistent with the significant increase in T cells and decrease seen in Tregs. In the untreated group, 16.7% (N = 1) developed active AMR whereas no animals in the APRIL/BLyS blockade treated group developed AMR. Significant tubulitis, interstitial inflammation, and vasculitis was noted in both untreated and treated transplant groups. Syngeneic transplant group did not develop any ACR or AMR (Table 1).

Renal function was measured in each group using a urine protein to creatinine (UPC) ratio. Consistent with the rejection seen on histology, untreated and treated transplant groups had worse renal function compared to syngeneic transplant, although not significantly different (Fig. 7D). In the syngeneic transplant group, 66.7% (N = 2) and 33.3% (N = 1) of animals had none (UPC <0.5) or mild proteinuria, respectively. APRIL/BLyS blockade treated group demonstrated slightly worse renal function than the untreated group. Seventy-five percent (N = 3) in the untreated group had no proteinuria compared to 33.3% (N = 1) in the treated group. No animals in the untreated group had moderate proteinuria (UPC 1.5–2.0), however 33.3% (N = 1) in the treated group developed moderate proteinuria (Fig 7D).

## Discussion

Chronic rejection remains a significant cause of late kidney allograft loss; therefore, novel therapies are required to treat this ongoing problem. Bm12 (H-2$^{bm12}$) to c57/B6 (H-2$^b$) minor MHC mismatch has previously been described as a murine chronic heart rejection model with severe cardiac allograft vasculopathy and fibrosis evident within 4 weeks.[27–29] However, this chronic rejection model has yet to be described in the kidney transplant literature. Therefore,

**Table 1. Banff scoring.**

| Banff score | Tx control (N = 3) Mean | SD | No treatment (N = 6) Mean | SD | APRIL/BLyS blockade (N = 6) Mean | SD | P |
|---|---|---|---|---|---|---|---|
| t | 0.3 | 0.6 | 3.0 | 0.0 | 3.0 | 0.0 | NS |
| i | 0.0 | 0.0 | 3.0 | 0.0 | 3.0 | 0.0 | NS |
| g | 0.0 | 0.0 | 0.2 | 0.4 | 0.0 | 0.0 | NS |
| ah | 0.0 | 0.0 | 0.0 | 0.0 | 0.0 | 0.0 | NS |
| v | 0.0 | 0.0 | 1.7 | 0.5 | 1.5 | 1.0 | NS |
| ptc | 0.0 | 0.0 | 0.3 | 0.8 | 0.0 | 0.0 | NS |
| cg | 0.0 | 0.0 | 0.0 | 0.0 | 0.0 | 0.0 | NS |
| ci | 0.0 | 0.0 | 0.0 | 0.0 | 0.0 | 0.0 | NS |
| ct | 0.0 | 0.0 | 0.0 | 0.0 | 0.0 | 0.0 | NS |
| cv | 0.0 | 0.0 | 0.0 | 0.0 | 0.0 | 0.0 | NS |
| mm | 0.0 | 0.0 | 0.0 | 0.0 | 0.0 | 0.0 | NS |
| mvi | 0.0 | 0.0 | 0.5 | 1.2 | 0.0 | 0.0 | |
| **AMR, % (N)** Active | 0% (0) | — | 16.7% (1) | — | 0% (0) | — | NS |
| **ACR** | | | | | | | |
| Negative | 100% (3) | — | 0% (0) | — | 0% (0) | — | NS |
| I | 0% (0) | — | 0% (0) | — | 16.7% (1) | — | NS |
| IIA | 0% (0) | — | 33.3% (2) | — | 33.3% (2) | — | NS |
| IIB | 0% (0) | — | 66.7% (4) | — | 33.3% (2) | — | NS |
| III | 0% (0) | — | 0% (0) | — | 16.7% (1) | — | NS |

we investigated the effect of targeting the B lymphocyte survival factors APRIL and BLyS on chronic antibody mediated rejection in a murine kidney transplant model. APRIL/BLyS blockade via TACI-Ig resulted in significant changes in both B and T lymphocyte populations. Specifically, follicular, T2 marginal zone, mature B lymphocytes and plasma cells were depleted with APRIL/BLyS blockade. These changes in mature B lymphocyte populations were further confirmed by the disruption of splenic germinal center architecture in treated animals. Importantly, animals treated with APRIL/BLyS blockade demonstrated a significant decrease in anti-nuclear autoantibody that was reduced to syngeneic transplant levels. This autoantibody decrease is consistent with the significant depletion of plasma cells seen in APRIL/BLyS blockade treated animals, which are the primary source of antibody production. Immature B lymphocyte populations such as newly formed and T1 B lymphocytes, which are less dependent on APRIL and BLyS for survival, were not altered with APRIL/BLyS blockade. In addition to not altering immature B lymphocytes, long-lived memory B cells were not depleted.

Despite depleted mature B lymphocytes, these changes did not result in any difference in rejection seen in untreated versus treated groups. Significant cellular rejection was seen in APRIL/BLyS blockade treated animals, which corresponds to increased effector T lymphocyte and decreased regulatory T lymphocyte populations. No AMR developed in APRIL/BLyS blockade treated group whereas 16.7% (N = 1) of untreated animals demonstrated active antibody mediated rejection on biopsy. No differences in renal function, as measured by urine protein to creatinine ratio, were seen between untreated and treated groups.

Interestingly, CD3$^+$CD4$^+$ and CD3$^+$CD8$^+$ T lymphocytes significantly increased compared to both untreated animals and syngeneic transplant control. The increase in effector T lymphocytes demonstrated here may be due to the depletion of regulatory B lymphocytes, which would normally downregulate inflammation and autoimmunity through IL-10.[30]

APRIL and BLyS have been extensively studied as therapeutic targets to reduce autoantibody in several preclinical models of autoimmune diseases including systemic lupus erythematous (SLE), IgA nephropathy, Sjorgren's syndrome and rheumatoid arthritis.[31–35] In a preclinical murine SLE model, dual APRIL and BLyS inhibition, but not BLyS blockade alone, reduced plasma cells and IgM levels.[36] Furthermore, APRIL/BLyS blockade with TACI-Fc completely prevented SLE disease progression including renal injury and arrested autoantibody production and onset of nephritis.[37] As indicated above, decreased B lymphocytes and autoantibody production through APRIL/BLyS blockade has been established in preclinical autoimmune disease models. However, the novelty of our current study is that we report decreased autoantibody production in a preclinical kidney transplant model using APRIL/BLyS blockade. Although changes in B lymphocyte populations and decreased autoantibody production did not translate to improved renal function or decreased rejection, these initial findings are important to note and may indicate modifications that need to be made to the chronic rejection model presented here. It is possible that if the post-transplant period was extended beyond 4 weeks, then significant differences in AMR would be seen between untreated and treated groups as was seen in circulating autoantibody levels. Additionally, future investigations will use APRIL/BLyS blockade in conjunction with T lymphocyte depleting agents to determine its efficacy in chronic rejection. By including a T lymphocyte depleting agent

## Conclusions

APRIL/BLyS blockade successfully reduced autoantibody production in a novel chronic kidney transplant rejection model. Decreased autoantibody production is likely due to depleted mature B lymphocyte populations including plasma cells. The findings presented here are further supported by the autoimmune literature in which APRIL/BLyS inhibition has been extensively studied as a method to reduce autoantibody and disease severity. However, this study remains novel due to the finding of decreased autoantibody with APRIL/BLyS blockade in a kidney transplant model. Additionally, the commonly used Bm12 to B6 chronic heart transplant model has not previously been described in kidney transplant literature. Despite these changes, cellular rejection and kidney function were not improved, which may be due to the absence of any T cell centric immunosuppression. Regardless, these findings suggest that APRIL/BLyS blockade may play a role in decreasing antibody formation long-term in kidney transplantation.

## Acknowledgments

The authors would like to thank the UWCCC Flow Cytometry Shared instrumentation core, including the Shared Instrumentation grant 1S00OD018202-01 Special BD LSR Fortessa, which made possible the purchase and use of the BD LSR Fortessa and the UW Transplant Research Training Grant (T32 AI125231). TACI-Ig was kindly provided by EMD Serono Research and Development Institute under an MTA.

## Author Contributions

**Conceptualization:** Natalie M. Bath, Robert R. Redfield III.

**Data curation:** Natalie M. Bath, Xiang Ding, Bret M. Verhoven, Nancy A. Wilson, Lauren Coons, Adarsh Sukhwal.

**Formal analysis:** Natalie M. Bath, Bret M. Verhoven, Weixiong Zhong, Robert R. Redfield III.

**Funding acquisition:** Robert R. Redfield III.

**Methodology:** Bret M. Verhoven, Nancy A. Wilson.

**Supervision:** Robert R. Redfield III.

**Writing – original draft:** Natalie M. Bath, Nancy A. Wilson, Robert R. Redfield III.

**Writing – review & editing:** Natalie M. Bath, Nancy A. Wilson, Weixiong Zhong, Robert R. Redfield III.

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
