## [Decision Letter · Decision Letter 0]

22 Aug 2019

PONE-D-19-21251

Autoantibody Production Significantly Decreased with APRIL/BLyS Blockade in Murine Chronic Rejection Kidney Transplant Model

PLOS ONE

Dear Dr. Redfield III,

Thank you for submitting your manuscript to PLOS ONE. After careful consideration, we feel that it has merit but does not fully meet PLOS ONE’s publication criteria as it currently stands. Therefore, we invite you to submit a revised version of the manuscript that addresses the points raised during the review process.

Please address our reviewers comments.

We would appreciate receiving your revised manuscript by Oct 06 2019 11:59PM. To enhance the reproducibility of your results, we recommend that if applicable you deposit your laboratory protocols in protocols.io, where a protocol can be assigned its own identifier (DOI) such that it can be cited independently in the future. For instructions see: http://journals.plos.org/plosone/s/submission-guidelines#loc-laboratory-protocols

We look forward to receiving your revised manuscript.

Kind regards,

Maria Lourdes Gonzalez Suarez, MD, PhD

Academic Editor

PLOS ONE

Journal Requirements:

Additional Editor Comments:

Decreased production of autoantibodies is an expected result of the therapy with APRIL/BLyS blockade. This study does help to direct efforts toward answering the question on whether therapy for AMR would have a direct effect on cellular mediated rejection in kidney transplantation. I agree with the authors that perhaps if follow up time was increased, it would be possible to see a difference in the treated versus untreated group. Study was well conducted.

Please address the comments of our reviewers.

Reviewers' comments:

Reviewer's Responses to Questions

**Comments to the Author**

1. Is the manuscript technically sound, and do the data support the conclusions?

Reviewer #1: Yes

Reviewer #2: Yes

Reviewer #3: Partly

2. Has the statistical analysis been performed appropriately and rigorously? 

Reviewer #1: Yes

Reviewer #2: Yes

Reviewer #3: Yes

3. Have the authors made all data underlying the findings in their manuscript fully available?

Reviewer #1: Yes

Reviewer #2: Yes

Reviewer #3: Yes

4. Is the manuscript presented in an intelligible fashion and written in standard English?

Reviewer #1: Yes

Reviewer #2: Yes

Reviewer #3: Yes

5. Review Comments to the Author

Reviewer #1: Relevant manuscript, well written and provided testing the use of APRIL/BLyS blockade as a potential therapeutic target to prevent allograft rejection in an in vivo mouse model of kidney transplant, comparing pathological findings with clinical measures.

Comments

1. Line 316: "Seventy-five percent (n=3) in the untretaed group had no proteinuria compared to 33.33% (n=1) in the treated group". Please change 33.33% to 25%, if in deed your total N=4.

2. Figure 1, panel C. Arrow is supposed to indicated negative staining, but the arrow is actually pointing to a green area of less intensity when compared to panel B. This could be still interpreted as a positive staining, although difference in intensity is noticeable. Is this auto-fluorescence?

Reviewer #2: This study done by Dr. N. M. Bath et al is a well done study. They have described outcomes of blockade of April/BlyS in a murine cAMR kidney transplant model, showing decrease in autoantibody production and altered splenic germinal center however it also shows there were no differences in kidney transplant pathology. The results from this study suggest that APRIL/BLyS blockade may have a role in decreasing antibody formation in long-term in kidney transplantation. Over all the methods applied are satisfactory and the figures represent mentioned results. Also this study enhances our knowledge about chronic anti body medicated rejection animal model in kidney transplant.

Reviewer #3: Overall, I found the manuscript interesting as the management of chronic antibody mediated rejection of the transplanted kidney remains a major challenge for practicing nephrologists and has a significant impact on the allograft longevity. The problem is a relevant one and newer/better therapies are much needed.

One of my concerns is about the novel chronic kidney rejection model used. The previously described vasculopathic changes from the heart transplant literature could not be reproduced in any of the transplant kidney biopsies. Along the same lines, no other histological changes consistent with chronic AMR or chronic active AMR were seen in the sensitized animals irrespective of the treatment arm.

The experiments were well described and consistent with previous publications by your group. The small sample size and short follow-up time added to the concerns about the model actually representing the disease certainly limit the conclusions.

6. PLOS authors have the option to publish the peer review history of their article (what does this mean?). If published, this will include your full peer review and any attached files.

Reviewer #1: No

Reviewer #2: Yes: Aditya Singh Pawar

Reviewer #3: No

---

## [Author Response · Author response to Decision Letter 0]

26 Sep 2019

Reviewer #1: Relevant manuscript, well written and provided testing the use of APRIL/BLyS blockade as a potential therapeutic target to prevent allograft rejection in an in vivo mouse model of kidney transplant, comparing pathological findings with clinical measures.

Comments

1. Line 316: "Seventy-five percent (n=3) in the untretaed group had no proteinuria compared to 33.33% (n=1) in the treated group". Please change 33.33% to 25%, if in deed your total N=4.

The numbers listed (lines 311-319) are correct. Unfortunately, we had very little urine collected on animals so we were not able to perform a urine protein:creatinine ratio (UPC) for every animal in this study. Number of urine specimens available for each group are as follows: syngeneic transplant (N=3); untreated transplant (N=4); treated transplant (N=3).

2. Figure 1, panel C. Arrow is supposed to indicated negative staining, but the arrow is actually pointing to a green area of less intensity when compared to panel B. This could be still interpreted as a positive staining, although difference in intensity is noticeable. Is this auto-fluorescence?

This is correct that the arrow in Figure 1C is pointing to an area that is green, although less intense than 1B. Figure 1A is truly negative (no green staining), whereas 1C is of lesser intensity (as also indicated in 1D with B6-B6 being near 0% and treated group being near 1%). We have changed the language to in the figure caption to refer to “lower intensity staining.” Changes made to lines 208-209.

Reviewer #2: This study done by Dr. N. M. Bath et al is a well done study. They have described outcomes of blockade of April/BlyS in a murine cAMR kidney transplant model, showing decrease in autoantibody production and altered splenic germinal center however it also shows there were no differences in kidney transplant pathology. The results from this study suggest that APRIL/BLyS blockade may have a role in decreasing antibody formation in long-term in kidney transplantation. Over all the methods applied are satisfactory and the figures represent mentioned results. Also this study enhances our knowledge about chronic anti body medicated rejection animal model in kidney transplant.

Reviewer #3: Overall, I found the manuscript interesting as the management of chronic antibody mediated rejection of the transplanted kidney remains a major challenge for practicing nephrologists and has a significant impact on the allograft longevity. The problem is a relevant one and newer/better therapies are much needed.

One of my concerns is about the novel chronic kidney rejection model used. The previously described vasculopathic changes from the heart transplant literature could not be reproduced in any of the transplant kidney biopsies. Along the same lines, no other histological changes consistent with chronic AMR or chronic active AMR were seen in the sensitized animals irrespective of the treatment arm.

The experiments were well described and consistent with previous publications by your group. The small sample size and short follow-up time added to the concerns about the model actually representing the disease certainly limit the conclusions.

We agree that the model presented here is not perfect due to the fact that chronic AMR was not noted in any sensitized animals. As stated (lines 377-379), if the post-transplant period had been extended beyond 4 weeks, the changes in ANA production and immune cell types may have also been seen in histology. In future models, the follow up time will likely need to be extended past 4 weeks. However, we believe the changes in ANA production and B cell populations seen from this model at 4 weeks are important findings.

---

## [Editor Report · Decision Letter 1]

2 Oct 2019

Autoantibody Production Significantly Decreased with APRIL/BLyS Blockade in Murine Chronic Rejection Kidney Transplant Model

PONE-D-19-21251R1

Dear Dr. Redfield III,

We are pleased to inform you that your manuscript has been judged scientifically suitable for publication and will be formally accepted for publication once it complies with all outstanding technical requirements.

With kind regards,

Maria Lourdes Gonzalez Suarez, MD, PhD

Academic Editor

PLOS ONE

Additional Editor Comments (optional):

Thank you for addressing reviewers comments. Manuscript is improved and ready for acceptance.
---

## [Editor Report · Acceptance letter]

10 Oct 2019

PONE-D-19-21251R1 

Autoantibody Production Significantly Decreased with APRIL/BLyS Blockade in Murine Chronic Rejection Kidney Transplant Model 

Dear Dr. Redfield III:

I am pleased to inform you that your manuscript has been deemed suitable for publication in PLOS ONE. Congratulations! Your manuscript is now with our production department. 

With kind regards,

on behalf of

Dr. Maria Lourdes Gonzalez Suarez 

Academic Editor

PLOS ONE